# What are young Indians saying about mental health? A content analysis of blogs on the It's Ok To Talk website

Pattie Pramila Gonsalves,[1] Eleanor Sara Hodgson,[1] Daniel Michelson,[2] Sweta Pal,[1] John Naslund,[3] Rhea Sharma,[1] Vikram Patel[3]

[1]Sangath, New Delhi, Delhi, India
[2]School of Psychology, University of Sussex, Brighton, UK
[3]Department of Global Health and Social Medicine, Harvard Medical School, Boston, Massachusetts, USA

**Correspondence to**
Pattie Pramila Gonsalves;
pattie.gonsalves@sangath.in

## ABSTRACT

**Objectives** This study used thematic content analysis to examine submissions to a youth mental health website, www.itsoktotalk.in, in India.

**Setting** We considered submissions made to the It's OK to Talk web platform during the first year of its operation (April 2017–March 2018), focusing specifically on website users based in India.

**Participants** We analysed 37 submissions by 33 authors aged 19–31 years (mean age 22 years) from seven Indian cities (New Delhi, Lucknow, Bengaluru, Mumbai, Pune, Hyderabad and Haryana). Eligible submissions were English-language first-person accounts of self-identified mental health problems, submitted in any media format for online publication by authors aged 18 years or older and who were based in India. Eight study participants were additionally involved in a focus group who contributed to the coding process and preparation of the final manuscript.

**Results** Four themes were identified: (1) living through difficulties; (2) mental health in context; (3) managing one's mental health and (4) breaking stigma and sharing hope. Overall, the participants expressed significant feelings of distress and hopelessness as a result of their mental health problems; many described the context of their difficulties as resulting from personal histories or wider societal factors; a general lack of understanding about mental health; and widespread stigma and other negative attitudes. Most participants expressed a desire to overcome mental health prejudice and discrimination.

**Conclusions** Personal narratives offer a window into young people's self-identified priorities and challenges related to mental health problems and recovery. Such insights can inform antistigma initiatives and other public awareness activities around youth mental health.

## INTRODUCTION

I hope that reading my story might help others feel less alone. It is important to know that other people are fighting the same battles. This is for anyone who feels stuck, overwhelmed or hopeless. You are not alone, just try to hold on. (Author, female, 24, www.itsoktotalk.in)

India has the world's largest population of young people aged 10–24,[1] and mental health problems are the leading health concern for

---

**Strengths and limitations of the study**

► Anonymised, first-person accounts of mental health problems and recovery offer the potential to illuminate issues that may not arise in other contexts (eg, clinical encounters).
► Participatory study design (involving youth participants in the coding process and manuscript preparation) may have benefits for the quality of research.
► Convenience sampling may have resulted in a narrow demographic cohort of self-selected, highly educated, English speakers.
► Narrative descriptions of mental health problems could not be differentiated into diagnostic or other clinical categories.

---

this group. The negative impacts of youth mental health problems are starkly reflected in the strong association between poor adolescent mental health and long-term social disability, while suicide is the leading cause of death for people aged 15–24 years in India.[2–4] There is growing interest in the use of social media as a vehicle for mental health promotion and stigma reduction at a population level, particularly among young people. This reflects the developmental significance of peer influence in the adolescent years and the rapid uptake of internet and mobile technology by young people across the globe.[1 5] Online social networks may also be harnessed to support recovery from mental problems.[6] Such opportunities are especially relevant in India where rapidly increasing numbers of young people are becoming connected to technology and social media.[7] At the same time, fewer than 10% of young Indians have access to formal mental health services, highlighting an urgent need to identify innovative strategies to promote mental health for this age group nationally and in low-income and middle-income countries more generally.[8 9]

## Sharing mental health stories online

Stories are essential as a means of how scientific knowledge, in its generality, can be applied to individuals in their particularity.[10]

'Social media' refers to websites and applications that enable users to create and share content or participate in social networking.[11] Social media platforms are inexpensive and relatively easy to use, enabling individuals to bring personal experiences into the public domain and thereby impact attitudes among larger groups.[12 13] Betton *et al*[12] argue that individuals, rather than institutions, are leading the way in bringing mental health conversations into online spaces, particularly through social media. Existing research highlights a complex picture of social media's relationship with health and well-being outcomes demonstrating both positive and negative impacts.[14] Though public discourse on social media use often emphasises negative impacts, some research suggests that the effects may depend on the way these platforms are used.[15] Individuals with stigmatised illnesses often avoid seeking healthcare.[16] For such groups, reading or hearing about the experiences of others can challenge stigma[17] and affect health-related decision-making and adjustment to an illness.[5] Young people with mental health difficulties report that one of the primary reasons for connecting with others online is to feel less alone.[18–20] Research also shows that young people may be more inclined to seek help for their mental health if they feel able to express their feelings, as well as being equipped with knowledge about mental health issues and available sources of support.[21]

## About 'It's Ok To Talk'

The It's Ok To Talk website (www.itsoktotalk.in)[22] was launched on WHO World Health Day (7 April 2017) as part of a public engagement initiative by the Indian NGO Sangath. It took inspiration from the theme for that year, 'Let's Talk', which emphasised the central role of disclosure in recovery from depression.[23] There is also evidence that disclosing a mental health problem can reduce self-stigma.[24 25] The website was designed in consultation with young people aged 15–24 years in New Delhi originally as an English-language social media platform for young people to share their stories of mental health problems and recovery. Authors were given the choice to submit any kind of media and publish anonymously or to include their name, location and gender. All submissions required an email address and optionally a phone number for administrative purposes.

## About this study

Comparatively, few studies have explored first-person accounts of the lived experience of youth mental health difficulties.[26] Such self-directed narratives may provide rich insights that are not available in other contexts (eg, clinical encounters).[27] The current study aimed to understand the first-hand experiences of mental health problems based on information shared by young Indians through the website www.itoktotalk.in. An important feature of the research was the participation of website users in the coding of qualitative data and preparation of the final manuscript. These activities ensured that the reported results would be grounded in the lived experience of young people.

## METHODS

### Setting and participants

The study's sampling frame included submissions made to the It's OK to Talk website during the first year of its operation (April 2017–March 2018). Eligible submissions were English-language, first-person accounts of self-identified mental health problems, submitted in any media format for online publication by authors aged 18–31 years and of Indian nationality. Forty-eight submissions from 44 authors met these criteria and were contacted for consent. An online consent form was sent to all authors by email and supplemented by a telephone call to review the study's aims, methods and data protection procedures. Thirty-three participants gave their consent; 11 authors did not respond to two email reminders and three phone call attempts.

### Ethical issues

Approval was obtained prior to commencement from the Sangath Institutional Review Board (IRB) for research with human subjects. Only submissions published on the website and therefore publicly accessible were considered for inclusion. This is in line with recommendations for the ethical conduct of internet research.[28 29] Personally identifiable data were removed and all direct quotations were paraphrased to ensure anonymity.

### Data collection

Instructions to prospective authors elicited submissions in 'any media form of your choice', along with further details (supported by FAQs) about language, file size, originality and anonymity. The instructions also requested that visual art submissions should be accompanied by text that contextualised the image(s). This written content was used as the basis for interpreting and coding the artwork submissions. All submissions were downloaded from the website, and those containing spoken text (eg, in audio or video submissions) were transcribed.

### Data analysis

We followed an integrated inductive–deductive approach to thematic content analysis.[30] Coders included two clinical psychologists (ESH and RS) and one public engagement specialist (PPG), with a post-doctoral researcher (JN) acting in the role of auditor; two coders were Indian nationals (PPG and RS). In the first instance, coders familiarised themselves with the full data set by reading and rereading transcripts. A random subset of five submissions was independently analysed line-by-line by each coder, with descriptive text-based codes assigned to the smallest

identifiable meaning units. This produced 33 initial codes which were compared; consensus was reached on a refined 'start list' of codes,[30] grouped into four themes with nine subthemes. The analysis then followed an iterative process in which coders independently applied the coding framework to 10 transcripts at a time, before reaching consensus on a revised framework through comparison and discussion. Differing views were resolved through group discussion with all authors. Group discussions also considered potential sources of personal bias in the interpretation of the data. To this end, coders maintained personal journals in which they reflected on the influence of their own gender, race, ethnicity, nationality, caste, class and professional backgrounds.

The auditor reviewed the emergent coding framework by scrutinising five coded transcripts at two time points, offering additional feedback on the nomenclature and hierarchical structure of codes. During the preparation of the manuscript, senior researchers (DM and VP) reviewed drafts and provided comments, which resulted in minor changes to the coding structure and descriptive labels. nVivo version 10 was used for organising and coding the data.

### Patient and public involvement

After four rounds of iteration, a focus group was held to provide quality assurance and comments on the coding framework. Participants were randomly selected from among the list of submissions originating from New Delhi where the researchers were based. Eight young people were invited by email, all of whom agreed to participate. The group reviewed the coding framework as applied to several text excerpts and provided suggestions about how to paraphrase direct quotes from selected blogs. The intention was to retain intended meaning while ensuring anonymity. Participants highlighted issues regarding terminology such as 'treatment', which was considered to have negative connotations (eg, framing mental health problems as an 'illness'). Coders subsequently produced two further iterations of the coding framework, which included changes informed by participant feedback such as renaming the 'treatment' theme as 'healthcare'. In addition, focus group participants independently paraphrased quotations to be used in the manuscript. They were also invited to review and comment on a draft of the study manuscript.

### RESULTS

Data were extracted from 37 submissions by 33 authors (13 male; 20 female) aged 19–31 years (mean age 22 years) from seven Indian cities (New Delhi, Lucknow, Bengaluru, Mumbai, Pune, Hyderabad and Haryana). These included 24 pieces of prose (in the form of a personal essay or 'blog'), nine poems, three pieces of visual art and one video. Four participants had two submissions each and these were treated as individual data files for analysis. The analysis established four themes: (1) living through difficulties; (2) mental health in context; (3) managing one's mental health and (4) breaking stigma and sharing hope. Corresponding subthemes and illustrative quotes are presented below under each theme, along with frequency counts reflecting coverage of subthemes across submissions.

### Theme 1: living through difficulties

This theme comprised three subthemes: powerful, painful thoughts and feelings, loneliness and isolation and mental health as a daily struggle.

#### Powerful, painful thoughts and feelings (29/37)

Most narratives included descriptions of intense distress, pain, suffering and hopelessness. These often emphasised the palpable embodied nature of negative thoughts and feelings as being physically overwhelming and burdensome. Five narratives explicitly referred to suicidal thoughts and three spoke explicitly about self-harm.

> It was the first time when I resorted to self-inflicting physical pain as a coping mechanism. The idea was overwhelming and scary but the mental trauma was too much to handle. I couldn't think clear and grabbed my blade. (Male, 23)

A number of narratives described experiences of gradual onset of mental health problems, as well as sudden moments of realisation after reaching a 'breaking point'. Many narratives described feeling powerless and without control and described additional fear, guilt, shame and blame experienced towards themselves as a result of their mental health or 'failure' to control thoughts and feelings. Many narratives spoke about a desperate desire to make sense of their experience, of asking 'why' and what this meant for their identities.

> Anxiety and depression felt like parts of me that would take over at different times. Seeing yourself as mentally unwell makes you feel helpless about how to handle it. (Male, 23)

#### Loneliness and isolation (20/37)

Many narratives spoke about intense feelings of isolation and loneliness.

> I am alone on a never ending road to nowhere. I am enthralling the darkness, which blinds, even the light. (Female, 24)

Some participants described the tension between their desire for connection while simultaneously wishing to isolate themselves. Many talked about difficulties conveying their mental health experiences to others due to a lack of trust or fear of rejection if they opened up.

> I have no idea how to talk about depression with my peers, colleagues, relatives or friends. There are times when I isolate myself completely, avoiding any communication. (Male, 25)

## Mental health as a daily struggle (15/37)

A sense of burden and day-to-day struggle with mental health problems was reflected in a number of the narratives. Some authors likened their experience to that of being in fight, war or battle while others referred to feeling trapped, imprisoned, shackled or in a dungeon. Narratives conveyed a strong sense of an ongoing effort and fatigue both in managing their mental health and keeping up with the tasks of daily life.

> I feel caged from within. It's exhausting to keep up with the struggle. I am afraid I might be losing the battle. A part of me is dying everyday. (Male, 25)

Many narratives spoke about putting on a mask or 'faking' their way through daily life. They talked about how sleeping, watching TV, smoking, binge eating or drinking became ways to fill a 'void' or achieve temporary relief, with some authors highlighting these were maladaptive coping methods which made them feel worse in the longer term.

### Theme 2: mental health in context

This theme comprised two subthemes: personal, social and cultural context' and knowledge, attitudes and stigma.

## Personal, social and cultural context (14/37)

Some young people's narratives described the personal context of their mental health difficulties in terms of child abuse or growing up with a parent with mental health problems, while others alluded to wider societal factors and influences. Several authors otherwise mentioned focal precipitating events such as bereavement or being in an abusive relationship.

> Somewhere in myself I did know we wouldn't get married owing to conservative families and different religious faiths. We still hoped though and went ahead anyway. Although it gave us a space to ourselves, it gave him the access to beat me up, and no one else knew. (Female, 30)

Some narratives spoke about academic and career pressures and high expectations to 'achieve' held by themselves, family members and other people around them. They spoke about how acknowledging one's own mental health needs was seen as personal weakness, particularly within a culture focused on 'productivity' and achievement.

> I felt guilty about not working hard. I felt if I took a small break, disconnected myself or working a little less, admitting that I felt exhausted, I would be letting people down. (Female, 28)

Some authors wrote about their own socioeconomic privilege, class, caste and education and how these factors have affected their ability to access support for their mental health problems.

> I have been fortunate enough to be able to access appropriate care for my mental health condition. I was lucky that I could take a break from work and give myself some time to get better. A majority of people struggling with mental health problems do not have this. (Female, 29)

## Knowledge, attitudes and stigma (18/37)

Multiple narratives described a general lack of knowledge or understanding about mental health and experiences of negative and stigmatising attitudes. Several authors identified how others' preconceived ideas about 'who has mental health', what it means to be a 'man' or about their own identities resulted in denial or dismissal of their experiences.

> People told me, 'just cheer up', not to use depression as an 'excuse' and that 'medicine is not the answer'. (Female, 25)

A number of participants expressed negative attitudes, shame and guilt towards their own mental health or diagnosis and described struggling to come to terms with their difficulties. For some self-stigma or perceived public stigma had acted as a barrier to seeking help.

> In the beginning, I wasn't ready to accept the fact that there was something wrong, something that would make me look imperfect. I just wanted to be 'normal', like everyone else in my college. (Female, 25)

Several authors highlighted the need for accurate information about mental health so that people can identify when they have a problem and seek appropriate help.

> Awareness and access to information needs to increase. Then many more people can identify problems they might be facing and get the right kind of help they need and in time. (Female, 29)

### Theme 3: managing one's mental health

This theme comprised three subthemes: psychosocial support, healthcare and self-care and recovery.

## Psychosocial support (16/37)

Many narratives spoke about the key roles played by friends and family in providing emotional and practical support. Conversely, some narratives described a lack of support from their networks as being an additional burden. The importance of being able to talk with others about their mental health, and to be heard and not to be judged, was expressed strongly across young people's narratives.

> It did not really help when my peers made no effort to show compassion towards me or understand what I was going through. (Male, 25)

> I began to truly realize the importance of having family and friends around me, people who understood

me. All I needed was someone to lend me an ear, give me the space to break down. (Male, 25)

Other authors described positive experiences of friends and family encouraging and enabling them to access professional help. Awareness and knowledge about mental health by family and friends was seen as connected to their willingness and ability to offer meaningful support.

It became my job to educate my family about my condition. They are nice people, they care about me but I could see through their helpless eyes, they were clueless about how to help me… (Female, 29)

### Healthcare (21/37)
Young people described varied experiences of mental healthcare, ranging from outpatient sessions with psychologists and/or psychiatrists to hospital admissions. Some narratives pointed to the effort and courage needed to access mental health services in the first place and that finding the right diagnosis or treatment takes time. Several authors described professional help as having been pivotal to their recovery.

Therapy taught me to notice my ways of thinking and responding to things. I learnt ways to break unhelpful patterns and to build on the helpful ones. (Male, 28)

A number of narratives described positive experiences of prescribed psychotropic medication.

When I started taking medication I felt calm for the first time in my life. It was like I had been stumbling around in the darkness and suddenly it was light. The voices I heard finally were quiet. (Female, 25)

Some participants who found medication effective nevertheless expressed a reluctance or dislike of taking medication and a desire not to do so in the future. Others described negative experiences of medication and of being medicated against their will.

My voices didn't like the meds. While they made my life hard, at least I felt something. Being dragged to the psychiatrist felt like a punishment when I had done nothing wrong. (Female, 23)

### Self-care and recovery (26/37)
Many narratives described experiences of recovery or improvements in mental health. Some participants shared experiences of self-care methods which helped them to maintain their well-being. These included taking care of their physical health; making changes to their relationship with work or study and finding ways to express themselves through creative activities or talking to others.

I have made changes in my office, making time in our meetings to talk about how we are feeling. It was a bit strange at first, but it soon became a vital part of

how we worked together and supported each other. (Female, 28)

Many narratives described a journey towards self-awareness and acceptance of themselves and their mental health difficulties as central to their recovery. Some authors conveyed the importance of holding on to hope through the difficult times and an acceptance that the future would contain ups and downs.

I still have my bad days but I am better at dealing with them now. True happiness still feels like a distant dream and thoughts about my future worry me. But I know I shall get there someday. (Female, 22)

Some authors note progress in relationships and achievements in their education or careers accomplished in spite of mental health difficulties. Others spoke about personal growth and learning achieved directly as a result of the experiences of mental health problems. Across the narratives authors described their identity as not being limited by their mental health or illness.

Experiencing a mental disorder taught me a lot about my own self. It made me grow and become stronger. It was the beginning of a new journey. I got back to what I loved doing, painting and writing! (Female, 19)

### Theme 4: breaking stigma and sharing hope
The majority of narratives contained messages directed at readers with mental health problems as well as a broader audience. Within these messages two subthemes were identified: desire to help others and 'desire to break stigma.

### Desire to help others (18/37)
Several authors shared the intention that their own story might help others by giving them hope, courage and reassurance that 'you are not alone'. Some authors spoke about positive experiences of sharing their mental health stories in their personal and professional lives and how helping others also helps them. Several authors gave specific advice to readers such as encouraging them to seek professional help. Across the narratives, there was a strong desire to convey that life can get better and to encourage readers not to 'give up'.

I hope I can help people to free themselves from being stuck in their own difficulties (Female, 20)

### Desire to break stigma (14/37)
Several authors identified their motivation for sharing their story in terms of normalising mental health problems and breaking stigma. Some narratives spoke about the importance of understanding mental health difficulties as a normal part of being human; others emphasised that mental health difficulties were 'real' and equally important as problems affecting physical health.

Mental illness is actually no different than a physical illness. It is just the same as having a cold, or a fracture or any physically manifested disease that you see a doctor for. (Female, 26)

The importance of mental health dialogue both personally and more broadly in society came out across many narratives Authors emphasised the courage required to disclose the state of ones' own mental health and stressed the powerful impact it can have on personal recovery and breaking wider stigma.

We must talk about mental health at every level—in schools, at work, with medical professionals and in the media. We need more information and awareness so that people can spot problems and get help. It's not just OK to talk, it is absolutely essential! (Female, 29)

## DISCUSSION

This study set out to explore the lived experiences of mental health problems and recovery from a sample of young people in India, based on mixed-media submissions to a public website. Four overarching themes were identified of (1) living through difficulties; (2) mental health in context; (3) managing one's mental health and (4) breaking stigma and sharing hope. Almost all submissions contained personal experiences of distress with the embodied experience of mental health difficulties as physically and mentally overwhelming and burdensome. Loneliness, isolation and a strong desire for connection were prominent, mirroring findings from qualitative studies of youth narratives obtained in high-income countries.[26 31]

The expression and sharing of such painful experiences appeared to be an important function of the It's Ok To Talk website. This is consistent with evidence that online spaces are commonly used by people with stigmatised health conditions for personal expression and social connection.[19 20] Informality and anonymity of the internet is emphasised in the extent to which it reduces the need to self-identify.[16] Youth consulted during the design of the It's Ok To Talk website also felt that most submitters might not wish to disclose their identities. However, although the website offered anonymity, most participants (89% in this study) chose to include their name, age and location. This could be a reflection of the strong desire to be heard and recognised.

Many participants described their mental health difficulties within the context of broader socioeconomic and cultural contexts. These submissions went beyond mere description of internal experiences to explore and challenge the societal and political contexts of psychological distress, offering a rich and multidimensional understanding of mental health. In particular, young people's narratives explained how rigid social norms can result in oppressive and unrealistic expectations, exacerbating

stress and contributing to poor mental health. In India, academic failure and stress have been noted in verbal autopsies of young people and as a predictor of suicidal ideation.[32] This is also consistent with research in other low-income and middle-income countries which has found academic, interpersonal and family difficulties to be among the key social determinants of poor mental health and suicide in young people.[4 33 34]

Participants raised concerns regarding equity of access to mental healthcare and acknowledged structural barriers that prevented the majority of Indians from finding or otherwise accessing care. Consistent with extensive previous research on the pervasive effects of mental health stigma on help-seeking, both public and self-stigma also limited young people's access to services.[35 36] The influence of stigma was prominent in narratives around recovery, and particularly the finding that acceptance of mental health problems appeared to be a central part of many participants' journeys of recovery. For some young people, sharing their story on the website was part of this process. This is also consistent with literature on the impact of disclosure on self-stigma.[25] Research has shown that selective online disclosure of potentially stigmatising health information may increase self-confidence, facilitate connections with similar others online and allow individuals to feel more accepting about their condition in offline settings.[6] In this study, in addition to fostering acceptance, many participants discussed other aspects of their lives and achievements and voiced that their mental health problems alone did not define them as a person.

Participants expressed mixed experiences of mental healthcare. In contrast to the bloggers studied by Marcus et al,[26] the young people in this study were largely positive about their experiences of mental health professionals and made recommendations to others about seeking professional help. Considering medication specifically, attitudes were mixed and predominantly negative. There is evidence of possible links between stigma and discontinuation of medication for mental health problems.[37–39] It would be important for future qualitative research to explore possible influences of stigma, fear of side effects and individual values on attitudes towards medication.

Participants variously expressed personal motivations to mitigate public stigma, to reduce self-stigma and to help others. For instance, to reassure others they are not alone in their struggle and normalise mental health as part of the human experience. The need for better information and understanding of mental health was seen as central to this process. Relatedly, participants emphasised the importance of broadening the mental health discourse to encompass social spaces ranging from homes, educational institutions, workplaces and community settings. Similarly, other research has indicated that online communities can serve as powerful venues for people to challenge stigma through personal empowerment and by providing hope to others.[17]

## Limitations

Limitations of this study included a convenience sample resulting in a narrow demographic cohort of self-selected, highly educated, English speakers (the website is in English) with access to the internet. Hence, participants are unlikely to be representative of the general youth population in India, which is a middle-income country with large disparities in internet access, especially among women and individuals from lower socioeconomic status.[40] This is also consistent with US data showing that individuals who share illness experiences online appear to be more highly educated than the general population.[41] A further limitation is that young people's mental health problems were self-reported and could not be differentiated into diagnostic or other clinical categories. This study also did not include any standardised measures to understand impacts on authors or readers of the website and findings do not tell us about possible benefits or risks of online narratives for authors or readers. For example, some research has identified risks associated with obtaining advice from peers with unknown credentials or setting of unrealistic expectations due to learning about others' health experiences.[42] Further research is needed to explore these important questions.

Notwithstanding these limitations, the study adds to the growing literature on digital communication and personal narratives, and how these can play a role in addressing the needs of young people living with mental health problems. Further efforts are required to determine whether similar online spaces can be used successfully by young people from lower socioeconomic groups, with lower education levels, from rural areas and from under-represented or marginalised population subgroups. The launch of a Hindi version of the It's Ok To Talk website is planned for 2019, with the aim to extend the reach of the site to include a large number of of young internet users in India whose primary computing language is Hindi.

## CONCLUSIONS

Social media offers opportunities to share personal narratives without temporal or spatial barriers.[12] Future research is needed to expand on the exploratory work described here and assess the effects of self-disclosure through online narratives. This is important both for promoting the mental health and well-being of those who have made these disclosures, and on attitudes regarding mental health problems in the wider community.

Such personal narratives highlight important gaps and needs in addressing young people's mental health needs. They also offer a window into young people's priorities in their own words and provide specific themes and nuanced key messages for awareness and help-seeking for future antistigma public engagement efforts in India or other similar settings.

Listen, don't try to solve the problem.

If you're struggling, know that it isn't your fault.

You need to educate yourself so you can help others.

There's no wrong time to get help.

The most important thing to remember is that you're not alone.

**Acknowledgements** The authors would like to thank all the participants who provided their valuable time to share their experiences and contribute to the coding and review of the manuscript. They are also grateful to Sunanda Jalote and Siddharth Parambi for their contributions.

**Contributors** PPG, ESH, DM, JN and VP designed the study with the support of SP and RS and PPG, ESH and RS coded the data. JN served as an auditor. PPG and ESH wrote the manuscript with support from DM, SP, JN, RS and VP. All authors contributed to interpretation of the data and gave final approval of the version to be published.

**Funding** This project has received funding from the Wellcome Trust, UK.

**Competing interests** None declared.

**Patient consent for publication** Not required.

**Ethics approval** Sangath Institutional Review Board (IRB).

**Provenance and peer review** Not commissioned; externally peer reviewed.

**Data sharing statement** Additional information is available by emailing pattie.gonsalves@sangath.in

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
