## [Reviewer comments · BMJ Open]

ARTICLE DETAILS

TITLE (PROVISIONAL)	What are young Indians saying about mental health? A content analysis of blogs on the It's Ok To Talk website
AUTHORS	Gonsalves, Pattie; Hodgson, Eleanor; Michelson, Daniel; Pal, Sweta; Naslund, John; Sharma, Rhea; Patel, Vikram

VERSION 1 - REVIEW

REVIEWER	Andrew Shepherd University of Manchester, UK
REVIEW RETURNED	16-Jan-2019

GENERAL COMMENTS	Dear Colleagues, Many thanks for the opportunity to review this fascinating manuscript addressing the early utilisation of an online resource for the sharing of narrative in relation to mental health. I agree that this represents an important area of investigation to better understand developing means of communication in relation to making sense of mental health whilst also affording an opportunity to develop novel means of addressing stigma. Overall, I felt that the manuscript was well written and easy to follow. My comments with regard to the submission relate particularly to the employed methodology: You identified 44 potential individuals who were approached to gain consent for their participation in the study with 33 offering consent. Is it possible to know any more information about the participants who refused consent? Particularly, any reasons offered for not participating? This may be of interest in shedding light on some of the issues faced in research in this area or a very real example of the perceived risk of stigma that you highlight in your findings? In your section on "Data collection" you suggest that some posts to the website involved artwork and that these were analysed together with any written or other contributions. I agree with your implication that the "multimedia" nature of online digital media is important and that images can serve an important role in terms of symbolism. However, I could not see from your methods precisely how these images (or other media) were incorporated into the analysis? What methods did you employ? Is it possible to provide any examples of the use of such media in your results & discussion sections? Perhaps in the form of a visual
---

	representation (if appropriate) or link to the media in question? Given the common contextual integration of other media alongside written contributions I felt that greater consideration of this domain could greatly enrich the analysis undertaken? You mention that three researchers and one auditor are involved in the coding process of your analysis. Please could you state how any discussion between researchers was handled and how any disagreements in the coding strategy were resolved? Your initial coding strategy states that you produced 33 codes grouped into 4 themes and 9 subthemes. How were themes evolved from the coding strategy? I felt that the manuscript could be strengthened through a greater consideration relating to issues of reflexivity in the analysis and reproduction of findings - that is, how did the research team and process impact on the interpretation of findings? You have stated the role of the coding team but not other researchers in the process? How were other members of the research team involved in the analysis and production of findings? How were issues of reflexivity considered in the analysis process? Further to the previous point, I was interested by your involvement of research participants in the analysis process. I felt, as you said, that this did represent an interesting contribution that moved the process beyond a simple content analysis. However, I felt that more information was required to explain how this approach was employed? For example, you state that the involvement of original contributing authors led to further iterations of the coding framework, beyond that produced by the research team. What changes did this participant involvement produce - can you give examples? Also, qualitative research is a technical process - how were competing views between the researchers and participants handled and discussed? Was any voice (form of expertise?) given priority over the other? In your discussion and summary section ("What do the new findings imply") you suggest that utilisation of online social media contributions may lead to authors / individuals experiencing a greater sense of connection or of being "heard." I did not feel that your data addressed these points directly as presented. There seemed to be an implication of a "desire" to be heard in terms of communicating normalising messages relating to the experience of mental health; however, these did feel to be more "desires" than direct experiences? I'm not certain that the experience of communicating via online means and feeling that one has been "heard" (witnessed?) is a simple one? Also, it is an experience that is likely to vary hugely depending on where one experiences one's self as being in terms of "recovery journey"? Are you able to expand on your findings to address these issues and consider these ideas / concerns? Kind regards
--	--

REVIEWER	John Goodwin University College Cork, Ireland
REVIEW RETURNED	28-Jan-2019

GENERAL COMMENTS	This is well-written paper, and the authors have taken a novel approach. However, there are issues around methodology and analysis: see comments below.
---

Abstract: The use of the phrase “thematic categories” is awkward. Themes represent latent content, while categories focus on manifest (surface) level data. It would be better to present these as “themes”, as this is what they appear to be.

Introduction: I have never seen a paper that started with a quote from participants. This works very well here!

However, the second quote (Brody 1987) doesn't really work where it is placed. It would be better not to have a quote introduce both the background and introduction sections.

Page 4, line 34: referring to social media as “it” is jarring, as “media” is a plural. Suggest: replacing “it” with “these platforms”, or something similar.

The introduction is very “pro social media”, and feels unbalanced. It would be important to address the problems that social media can cause in young people also.

Methods: It is not clear why an Indian blog would contain English language messages – why were people deliberately writing in a language that was not their first language? Would this not have affected interpretation, or the depth of their communication? Some clarity is needed here.

How were the authors of the blogs contacted? Did they all provide contact details? Similarly, do the authors know where each individual was from? Did all bloggers provide their geographical location? (They may have done, but this needs to be explicit, otherwise it raises questions). (this is answered later, but it needs to be stated earlier).

While I understand that the authors wish to protect anonymity of the participants, I would be concerned about the paraphrasing of direct quotations, as there is the potential for mis-interpretation here. Although some participants were involved in analysis, not all participants were, and so some quotes may be taken out of context.

The authors list partial participant-analysis as a strength, however, the authors should also consider reporting on the limitations of this approach. Identifying thematic content (latent data) should be objective, but participants cannot be objective if they are commenting on their own statements. If the aim of the paper was to present a manifest content analysis, then this would not be problematic; however, a thematic approach to content analysis requires much more objectivity.

Results: The authors provide some descriptive quantitative data, and this appropriate in some places, but not entirely appropriate in others. For example, where the authors state, “Most narratives (29/37) included descriptions of intense distress”, this is useful, as it provides the reader with context, and illustrates the frequency of certain topics. However, the authors state “This theme was the most dominant across the submissions (33/37)”, and this is not contextual, as this is a “theme”, and not just the reporting of manifest data or descriptive information. In any sort of thematic analysis (thematic analysis or thematic approach to qualitative

	content analysis), it is not recommended to quantify latent data in this fashion. There is inconsistency in the reporting of numerical data, with the authors stating “many” or “some” without numerical illustration. Because the authors provided several instances of such numerical data, it stands out when they then talk about “many” but do not provide numbers, a problem addressed in Sandeloski’s 2001 paper (Real qualitative researchers do not count: The use of numbers in qualitative research.). It may be best to drop all the numerical reporting – most qualitative researchers would not consider it standard to quantify qualitative data, anyway. Page 7, line 18: The authors refer to a sub-theme, “day-to-day struggles”. However, neither the content nor the quotes are about day-to-day struggles; thematically, this is more about inability to express emotions. Both quotes refer to “bottling up” or “being caged” Page 8, line 30: as the authors refer to self-stigma, they should also refer to “fear of stigma” as “perceived public stigma” Overall, the analysis needs more depth rather than breadth. In several cases, particularly under the third theme, the authors provide one line, a quote, followed by another line, and another quote. A thematic analysis requires much more interpretation. It may be the case that some sub-themes can be conflated. Discussion: The authors talk about participants’ comments about physical health: this was not present in the analysis – there was no discussion about the impact of mental health on physical health. Page 20, line 51: the word “categories” is used here, but these are previously referred to as sub-themes. Page 11, line 7: This answers my previous question about identification. However, it would be important to address these issues earlier, rather than appearing for the first time in the discussion. Overall, I very much enjoyed reading this paper, and, with some re-writing, I can see its potential contributions to the body of knowledge on subjective experiences of mental health, and on the merits of online approaches.
--	---

VERSION 1 – AUTHOR RESPONSE

REVIEWER COMMENTS:

REVIEWER 1: ANDREW SHEPHERD

Dear Colleagues,

1) Many thanks for the opportunity to review this fascinating manuscript addressing the early utilisation of an online resource for the sharing of narrative in relation to mental health. I agree that this represents an important area of investigation to better understand developing means of communication in relation to making sense of mental health whilst also affording an opportunity to develop novel means of addressing stigma.

We thank the reviewer for their positive feedback.

2) Overall, I felt that the manuscript was well written and easy to follow. My comments with regard to the submission relate particularly to the employed methodology:

You identified 44 potential individuals who were approached to gain consent for their participation in the study with 33 offering consent. Is it possible to know any more information about the participants who refused consent? Particularly, any reasons offered for not participating? This may be of interest in shedding light on some of the issues faced in research in this area or a very real example of the perceived risk of stigma that you highlight in your findings?

We thank the reviewer for these questions. Of the 44 individuals contacted (via email and a telephone call), 33 participants responded and gave consent for participation and 11 individuals did not respond to two email reminders or three attempted phone calls. Of the participants who responded, all provided consent. We have updated the 'Setting and participants' subsection of the Methods on page 4 to include these details, as below:

Setting and participants

The study's sampling frame included submissions made to the It's OK to Talk website during the first year of its operation (April 2017 - March 2018). Submissions required an email address and optionally a phone number for administrative purposes. Eligible submissions were English language, first-person accounts of self-identified mental health problems, submitted in any media format for online publication by authors aged 18-31 years and of Indian nationality. Forty-eight submissions from 44 authors met these criteria and were contacted for consent. An online consent form was sent to all authors by email and supplemented by a telephone call to review the study's aims, methods and data protection procedures. Thirty-three participants gave their consent; 11 authors did not respond to two email reminders and three phone call attempts.

3) In your section on "Data collection" you suggest that some posts to the website involved artwork and that these were analysed together with any written or other contributions. I agree with your implication that the "multimedia" nature of online digital media is important and that images can serve an important role in terms of symbolism. However, I could not see from your methods precisely how these images (or other media) were incorporated into the analysis? What methods did you employ? Is it possible to provide any examples of the use of such media in your results & discussion sections? Perhaps in the form of a visual representation (if appropriate) or link to the media in

question? Given the common contextual integration of other media alongside written contributions I felt that greater consideration of this domain could greatly enrich the analysis undertaken?

We thank the reviewer for this question. Further details about the analysis of visual media have been added to the 'Data collection' subsection in the Methods section on page 4, as below:

The instructions also requested that visual art submissions should be accompanied by text that contextualised the image(s). This written content was used as the basis for interpreting and coding the artwork submissions. All submissions were downloaded from the website, and those containing spoken text (e.g. in audio or video submissions) were transcribed.

We agree that including images in the manuscript would add richness to our reporting of the results. However this would not be possible without making the author identifiable, as it could be image searched; we therefore decided not to include any images in the manuscript.

4) You mention that three researchers and one auditor are involved in the coding process of your analysis. Please could you state how any discussion between researchers was handled and how any disagreements in the coding strategy were resolved?

Further details have been added to the 'Data analysis' subsection in the Methods section on page 4, as below:

Disagreements were resolved through group discussion; had this not been possible, then decisions would have followed the majority view. Group discussions also considered potential sources of personal bias in the interpretation of the data. To this end, coders maintained personal journals in which they reflected on the influence of their own gender, race, ethnicity, nationality, caste, class and professional backgrounds.

5) Your initial coding strategy states that you produced 33 codes grouped into 4 themes and 9 subthemes. How were themes evolved from the coding strategy?

Following the integrated inductive-deductive method the initial list of codes was informed by the coders' knowledge of the literature and professional knowledge, their familiarity with the overall data set as well as the specifics of the detailed line-by-line coding of a subset of articles. The intention of the initial list was to map out the likely territory of themes. Themes were arrived at through a process of grouping similar and related codes together into sub-themes, which were in turn grouped into overarching (higher-order) themes. These groupings were created through a mapping exercise and

discussion between the 3 coders. Again, this was informed by the coders' prior knowledge of the literature and of mental health and psychology. We have added additional details to the 'Data analysis' subsection in the Methods on page 4 and 5 to clarify the above.

We followed an integrated inductive-deductive approach to thematic content analysis[29]. Coders included two clinical psychologists (ESH and RS) and one public engagement specialist (PPG), with a postdoctoral researcher (JN) acting in the role of auditor; two coders were Indian nationals (PPG and RS). In the first instance, coders familiarised themselves with the full data set by reading and re-reading transcripts. A random subset of five submissions was independently analysed line-by-line by each coder, with descriptive text-based codes assigned to the smallest identifiable meaning units. This produced 33 initial codes which were compared; consensus was eventually reached on a refined 'start list' of codes[29], grouped into 4 themes with 9 sub-themes. The analysis then followed an iterative process in which coders independently applied the coding framework to 10 transcripts at a time, before reaching consensus on a revised framework through comparison and discussion. Disagreements were resolved through group discussion; had this not been possible, then decisions would have followed the majority view. Group discussions also considered potential sources of personal bias in the interpretation of the data. To this end, coders maintained personal journals in which they reflected on the influence of their own gender, race, ethnicity, nationality, caste, class and professional backgrounds.

The auditor reviewed the emergent coding framework by scrutinising five coded transcripts at two time points, offering additional feedback on the nomenclature and hierarchical structure of codes. During the preparation of the manuscript, senior researchers (DM, VP) reviewed drafts and provided comments, which resulted in minor changes to the coding structure and descriptive labels.

6) I felt that the manuscript could be strengthened through a greater consideration relating to issues of reflexivity in the analysis and reproduction of findings - that is, how did the research team and process impact on the interpretation of findings? You have stated the role of the coding team but not other researchers in the process? How were other members of the research team involved in the analysis and production of findings? How were issues of reflexivity considered in the analysis process?

We have updated the 'Data analysis' subsection of the Methods on page 4 to include more details on these aspects.

Group discussions also considered potential sources of personal bias in the interpretation of the data. To this end, coders maintained personal journals in which they reflected on the influence of their own gender, race, ethnicity, nationality, caste, class and professional backgrounds. The auditor reviewed the emergent coding framework by scrutinising five coded transcripts at two time points, offering additional feedback on the nomenclature and hierarchical structure of codes. During the preparation of the manuscript, senior researchers (DM, VP) reviewed drafts and provided comments which resulted in minor changes to the coding structure and descriptive labels.

7) Further to the previous point, I was interested by your involvement of research participants in the analysis process. I felt, as you said, that this did represent an interesting contribution that moved the process beyond a simple content analysis. However, I felt that more information was required to explain how this approach was employed? For example, you state that the involvement of original contributing authors led to further iterations of the coding framework, beyond that produced by the research team. What changes did this participant involvement produce - can you give examples? Also, qualitative research is a technical process - how were competing views between the researchers and participants handled and discussed? Was any voice (form of expertise?) given priority over the other?

We have revised the 'Patient and public involvement' subsection of the Methods on page 5 to include more details on these aspects.

After four rounds of iteration, a focus group was held to provide quality assurance and comments on the coding framework. Participants were randomly selected from among the list of submissions originating from New Delhi where the researchers were based. Eight young people were invited by email, all of whom agreed to participate. The group reviewed the coding framework as applied to several text excerpts and provided suggestions about how to paraphrase direct quotes from selected blogs. The intention was to retain intended meaning while ensuring anonymity. Participants highlighted issues regarding terminology such as "treatment", which was considered to have negative connotations (e.g. framing mental health problems as an "illness"). Coders subsequently produced two further iterations of the coding framework, which included changes informed by participant feedback such as renaming the "treatment" theme as "healthcare". In addition, focus group participants independently paraphrased quotations to be used in the manuscript. They were also invited to review and comment on a draft of the study manuscript.

8) In your discussion and summary section ("What do the new findings imply") you suggest that utilisation of online social media contributions may lead to authors / individuals experiencing a greater sense of connection or of being "heard." I did not feel that your data addressed these points directly as presented. There seemed to be an implication of a "desire" to be heard in terms of communicating normalising messages relating to the experience of mental health; however, these did feel to be more "desires" than direct experiences? I'm not certain that the experience of communicating via online means and feeling that one has been "heard" (witnessed?) is a simple one? Also, it is an experience that is likely to vary hugely depending on where one experiences one's self as being in terms of "recovery journey"? Are you able to expand on your findings to address these issues and consider these ideas / concerns?

We thank the reviewer for this important comment and question. Reflecting on this feedback we have made revisions to the fourth theme to highlight motivations to help others and to break stigma. The Summary on page 2 and Discussion on page 10 sections have also been revised. Excerpts of revised text are below and full revisions are reflected in updated manuscript.

FROM SUMMARY:

Strengths

- anonymised, first-person accounts of mental health problems and recovery offer the potential to illuminate issues that may not arise in other contexts (e.g. clinical encounters)

FROM DISCUSSION:

Participants variously expressed personal motivations to mitigate public stigma, to reduce self-stigma and to help others. For instance, to reassure others they are not alone in their struggle and normalise mental health as part of the human experience. The need for better information and understanding of mental health was seen as central to this process. Relatedly, participants emphasised the importance of broadening the mental health discourse to encompass social spaces ranging from homes, educational institutions, workplaces and community settings. Similarly, other research has indicated a growing acceptance that online communities can serve as powerful venues for people to challenge stigma through personal empowerment and by providing hope to others[15].

REVIEWER 2: JOHN GOODWIN

- 1) This is well-written paper, and the authors have taken a novel approach.

We thank the reviewer for this positive feedback.

- 2) However, there are issues around methodology and analysis: see comments below.

Abstract: The use of the phrase “thematic categories” is awkward. Themes represent latent content, while categories focus on manifest (surface) level data. It would be better to present these as “themes”, as this is what they appear to be.

We thank the reviewer for this helpful comment and have revised the manuscript to reflect this change from “thematic categories” to “themes”.

3) Introduction: I have never seen a paper that started with a quote from participants. This works very well here!

We thank the reviewer for this positive feedback.

4) However, the second quote (Brody 1987) doesn't really work where it is placed. It would be better not to have a quote introduce both the background and introduction sections.

We thank the reviewer for this feedback. The purpose of this placing this quote here was to highlight the importance and value of personal accounts of mental health to understand the interaction of mental health experiences, illness, suffering and social and cultural context. We are happy to leave the final decision on the inclusion of this quote to the editors.

5) Page 4, line 34: referring to social media as "it" is jarring, as "media" is a plural. Suggest: replacing "it" with "these platforms", or something similar.

We have revised the manuscript to reflect this change.

6) The introduction is very "pro social media", and feels unbalanced. It would be important to address the problems that social media can cause in young people also.

We thank the reviewer for this feedback and have revised the Introduction and 'Limitations' sub section of the Discussion on page 10 and 11 to be more neutral in tone and acknowledge some of the potential risks. An excerpt from the 'Limitations' is below and full revisions are reflected in updated manuscript.

A further limitation is that young people's mental health problems were self-reported and could not be differentiated into diagnostic or other clinical categories. This study also did not include any standardised measures to understand impacts on authors or readers of the website and findings do not tell us about possible benefits or risks of online narratives for authors or readers. For example, some research has identified risks associated with obtaining advice from peers with unknown credentials or setting of unrealistic expectations due to learning about others' health experiences[41]. Further research is needed to explore these important questions.

7) Methods: It is not clear why an Indian blog would contain English language messages – why were people deliberately writing in a language that was not their first language? Would this not have affected interpretation, or the depth of their communication? Some clarity is needed here.

It's Ok To Talk website was initially launched in English as India has over 175 million English internet users (KMPG 2017) and many young Indian social media users' primary/first language is English. We therefore do not expect that this would have negatively affected interpretation of their submissions. We have acknowledged the limitations of this narrow sample in the Discussion section of the paper.

As regional language internet and social media users in India are growing in number the website is also being launched in Hindi.

We have updated the 'About It's Ok To Talk' subsection of the Introduction on page 3 and 'Limitations' subsection of the Discussion on page 11 to highlight this better.

FROM INTRODUCTION:

The website was designed in consultation with young people aged 15-24 years in New Delhi originally as an English-language social media platform for young people to share their stories of mental health problems and recovery. Authors were given the choice to submit any kind of media and publish anonymously or to include their name, location and gender.

FROM DISCUSSION:

The launch of a Hindi version of the website is planned for 2019, with the aim to extend the reach of the site to include the majority of young internet users in India whose primary computing language is Hindi.

8) How were the authors of the blogs contacted? Did they all provide contact details? Similarly, do the authors know where each individual was from? Did all bloggers provide their geographical location? (They may have done, but this needs to be explicit, otherwise it raises questions). (this is answered later, but it needs to be stated earlier).

We thank the reviewer for this question. When authors made a submission on the website they were prompted to fill in their name, age, gender, email address, phone number and location and could choose whether to publish anonymously or not. Yes, authors know where each individual was from.

We have updated the 'Setting and participants' subsection of the Methods on page 4 to provide these details more explicitly. Please see our response to Reviewer 1 Comment 2.

9) While I understand that the authors wish to protect anonymity of the participants, I would be concerned about the paraphrasing of direct quotations, as there is the potential for mis-interpretation here. Although some participants were involved in analysis, not all participants were, and so some quotes may be taken out of context.

We thank the reviewer for this comment. We have revised the 'Patient and public involvement' sub section of the Methods on page 5 to make the paraphrasing process much more explicit. Please see our response to Reviewer 1 Comment 7. 10) The authors list partial participant-analysis as a strength, however, the authors should also consider reporting on the limitations of this approach. Identifying thematic content (latent data) should be objective, but participants cannot be objective if they are commenting on their own statements. If the aim of the paper was to present a manifest content analysis, then this would not be problematic; however, a thematic approach to content analysis requires much more objectivity.

We thank the reviewer for this comment. We agree that a content analysis requires objectivity and that this may be more difficult for participants who contributed their own stories. However, our process of participant involvement did not expect or require them to be objective. Although focus group participants were invited to comment on and input into the analysis process, this was very much taken as further data for the research team to analyze. We therefore do not feel that this is a limitation that requires reporting in the manuscript, however we have updated the "Patient and public involvement" sub-section of the Methods to further elaborate on the details of the focus group and be clearer about the role of the participants and the researchers. Please see our response to Reviewer 1 Comment 7.

11) Results: The authors provide some descriptive quantitative data, and this appropriate in some places, but not entirely appropriate in others. For example, where the authors state, "Most narratives (29/37) include descriptions of intense distress", this is useful, as it provides the reader with context, and illustrates the frequency of certain topics. However, the authors state "This theme was the most dominant across the submissions (33/37)", and this is not contextual, as this is a "theme", and not just the reporting of manifest data or descriptive information. In any sort of thematic analysis (thematic analysis or thematic approach to qualitative content analysis), it is not recommended to quantify latent data in this fashion. There is inconsistency in the reporting of numerical data, with the authors stating "many" or "some" without numerical illustration. Because the authors provided several instances of such numerical data, it stands out when they then talk about "many" but do not provide numbers, a problem addressed in Sandeloski's 2001 paper (Real qualitative researchers do not count: The use of numbers in qualitative research.). It may be best to drop all the numerical reporting – most qualitative researchers would not consider it standard to quantify qualitative data, anyway.

We thank the reviewer for this feedback and the interesting reference. This was indeed a debate that we had internally when writing the paper. We have revised our reporting of numbers in the manuscript removing numbers for themes and only reporting on sub-themes to illustrate the emphasis placed on the various sub-themes by the participants. We hope that the reviewers will agree that our level of reporting is helpful and appropriate to the analysis. To illustrate this, we have provided an example from Theme 1: Living through difficulties on page 5 manuscript below:

Theme 1: Living through difficulties

This theme was comprised of three sub-themes: “Powerful, painful thoughts and feelings”; “Loneliness and isolation”; and “Mental health as a daily struggle”.

Powerful, painful thoughts and feelings (29/37)

Most narratives included descriptions of intense distress, pain, suffering and hopelessness. These often emphasized the palpable embodied nature of negative thoughts and feelings as being physically overwhelming and burdensome. Five narratives explicitly referred to suicidal thoughts and three spoke explicitly about self-harm.

13) Page 7, line 18: The authors refer to a sub-theme, “day-to-day struggles”. However, neither the content nor the quotes are about day-to-day struggles; thematically, this is more about inability to express emotions. Both quotes refer to “bottling up” or “being caged”

We thank the reviewer for this helpful point. Whilst inability to express emotions is a significant component of this sub-theme, there are other aspects, such as the efforts to keep up with day-to-day life, which make the sub-theme broader but may not have been fully reflected in the previous results. Having re-examined the full content within this sub-theme we have revised the label as “Mental health as a daily struggle” and revised the manuscript to provide more detail on the other aspects of this sub-theme.

14) Page 8, line 30: as the authors refer to self-stigma, they should also refer to “fear of stigma” as “perceived public stigma”

We have updated the manuscript to reflect this.

15) Overall, the analysis needs more depth rather than breadth. In several cases, particularly under the third theme, the authors provide one line, a quote, followed by another line, and another

quote. A thematic analysis requires much more interpretation. It may be the case that some sub-themes can be conflated.

We thank the reviewer for this important feedback. We have reviewed the Results section and, in response to the suggestion for more depth, significantly revised the third and fourth themes. Within the third theme we have collapsed the sub-themes of “Self-care,” “Moving forward” and “Growth and hope” into a single sub-theme of “Selfcare and recovery”. Within the fourth theme, we have revised and renamed the subthemes from “It’s only human” and “Importance of talking and listening” to “Desire to help others” and “Desire to break stigma”. We hope that the reviewers will find the analysis significantly improved by this reworking.

16) Discussion: The authors talk about participants’ comments about physical health: this was not present in the analysis – there was no discussion about the impact of mental health on physical health.

We believe that the reviewer is making reference to the following sentence from the discussion: “Almost all submissions contained personal experiences of distress with the embodied experience of mental health difficulties as physically and mentally overwhelming and burdensome”. This is referring to the physicality of the experience of mental health mentioned in Theme 1 (“powerful, painful thoughts and feelings”) (on Page 6) rather than to physical health specifically.

Physical health is mentioned in the “Self-care and recovery” sub-theme in the Results section but this is not mentioned specifically in the Discussion. We hope that this clarifies the reviewer’s comment.

17) Page 20, line 51: the word “categories” is used here, but these are previously referred to as sub-themes.

We have revised the manuscript to reflect the change to “themes”.

18) Page 11, line 7: This answers my previous question about identification. However, it would be important to address these issues earlier, rather than appearing for the first time in the discussion.

We have updated the “About It’s Ok To Talk” sub section of the Introduction on page 3 and “Setting and participants” sub-section of the Methods on page 4 to ensure these details are clearly stated from the outset. Please refer to the response to Comment 7 and Reviewer 1 Comment 2 for the updated manuscript text.

19) Overall, I very much enjoyed reading this paper, and, with some re-writing, I can see its potential contributions to the body of knowledge on subjective experiences of mental health, and on the merits of online approaches.

Thank you. We hope that you agree that the manuscript is improved based on your helpful feedback.

VERSION 2 – REVIEW

REVIEWER	Andrew Shepherd University of Manchester, UK
REVIEW RETURNED	22-Mar-2019

GENERAL COMMENTS	Dear Colleagues, Many thanks for the opportunity to read this interesting manuscript for a second time. I also welcome the changes that have been made in response to previous reviewer comments. I am satisfied that the points I raised have been addressed to an appropriate degree. I would be happy to support its publication on this basis. Best wishes
---

REVIEWER	John Goodwin School of Nursing and Midwifery Brookfield Health Sciences Complex University College Cork Cork Ireland
REVIEW RETURNED	15-Mar-2019

GENERAL COMMENTS	This is a stronger draft. In particular, the results read much better. My main concern in the paraphrasing of quotations. I still feel the quote (“Stories are essential as a means of how scientific knowledge, in its generality, can be applied to individuals in their particularity”) feels out of place here – it weakens the impact of the first quote. As the reviewers have suggested, I will leave the decision to delete with the editor. I still feel that a more balanced perspective on social media needs to be included in the introduction. All this would take is one line, e.g. “although social media has been linked with increased mental distress in adolescents, it has the potential...” etc. To present social media as overly positive is to ignore the “elephant in the room”. I still have problems with the paraphrasing of quotes. I feel strongly about not paraphrasing data, as there is the possibility that the original meaning may be lost or misconstrued. Although I understand the rationale behind this, the platform was publicly available, and so these quotations would be accessible from the
--

	internet anyway. It does not make any sense to paraphrase these. Even the inclusion of participants in analysis does not justify this approach. Please present the original quotations. Unclear about this sentence: "Disagreements were resolved through group discussion; however, had this not been possible, then decisions would have followed the majority view". Were there disagreements? This sentence starts in the past tense (were resolved) but then moves into the subjunctive/hypothetical (had this not been possible). Although as peer reviewer I do not normally comment on editorial changes, please change phrases such as "this theme was comprised of two sub-themes" to either "this theme comprised two sub-themes" or "this theme was composed of two sub-themes". Almost every paragraph of the results starts with a quasi-quantitative statement, such as "some of the authors", "most of the authors", etc. As per my previous review, this feels like a quantitative approach to qualitative data analysis. It would read better to change this approach, to reflect a more qualitative ethos. The section "psychosocial support" talks about where support was provided and not provided. However, the quotes only seem to present the positive aspects here (i.e. where support was evident). For example it is not clear from this quote ("All I needed was someone to lend me an ear, give me the space to break down.") if the participants received support or not. Quotes that explicitly reflect both perspectives and experiences are warranted. (This is different to the following quote, where friends/family tried to offer support but just didn't know how). A more meaningful heading than "healthcare" is recommended.
--	--

VERSION 2 – AUTHOR RESPONSE

REVIEWER COMMENTS:

REVIEWER 1: ANDREW SHEPHERD

Many thanks for the opportunity to read this interesting manuscript for a second time. I also welcome the changes that have been made in response to previous reviewer comments. I am satisfied that the points I raised have been addressed to an appropriate degree. I would be happy to support its publication on this basis.

We thank the reviewer for this positive feedback and their time reading the manuscript again.

REVIEWER 2: JOHN GOODWIN

1) I still feel that a more balanced perspective on social media needs to be included in the introduction. All this would take is one line, e.g. “although social media has been linked with increased mental distress in adolescents, it has the potential...” etc. To present social media as overly positive is to ignore the “elephant in the room”.

The manuscript has been updated to reflect this. We have revised the “Sharing mental health stories online” subsection of the Introduction on page 3. We hope that the reviewer will find that our revision addresses this comment.

Existing research highlights a complex picture of social media's relationship with health and wellbeing outcomes demonstrating both positive and negative impacts[14]. Though public discourse on social media use often emphasizes negative impacts, some research suggests that the effects may depend on the way these platforms are used [15].

2) I still have problems with the paraphrasing of quotes. I feel strongly about not paraphrasing data, as there is the possibility that the original meaning may be lost or misconstrued. Although I understand the rationale behind this, the platform was publicly available, and so these quotations would be accessible from the internet anyway. It does not make any sense to paraphrase these. Even the inclusion of participants in analysis does not justify this approach. Please present the original quotations.

The paraphrasing of quotes is related to following best practice as this is a sensitive and personal topic, and stories are submitted to this platform on the understanding that it is a safe space to share. We revised the ‘Patient and public involvement’ sub section of the Methods on page 5 to make the paraphrasing process much more explicit for a reader.

Paraphrasing of quotes was also a condition of ethical approval and participant consent obtained for the study. We are happy to leave the final decision on the inclusion of participant quotes to the editor.

3) Unclear about this sentence: “Disagreements were resolved through group discussion; however, had this not been possible, then decisions would have followed the majority view”. Were there disagreements? This sentence starts in the past tense (were resolved) but then moves into the subjunctive/hypothetical (had this not been possible).

We have revised the section on Data Analysis on page 4 to make this clearer. A few differing views on codes were expressed by coders during the iterative process of coding however these were fairly easily resolved through group discussion amongst the coders and senior authors. We have revised the manuscript to make this clearer as below:

The analysis then followed an iterative process in which coders independently applied the coding framework to 10 transcripts at a time, before reaching consensus on a revised framework through comparison and discussion. Differing views were resolved through group discussion with all authors. Group discussions also considered potential sources of personal bias in the interpretation of the data. To this end, coders maintained personal journals in which they reflected on the influence of their own gender, race, ethnicity, nationality, caste, class and professional backgrounds.

4) Although as peer reviewer I do not normally comment on editorial changes, please change phrases such as “this theme was comprised of two sub-themes” to either “this theme comprised two sub-themes” or “this theme was composed of two sub-themes”.

These revisions have been made to the manuscript.

5) Almost every paragraph of the results starts with a quasi-quantitative statement, such as “some of the authors”, “most of the authors”, etc. As per my previous review, this feels like a quantitative approach to qualitative data analysis. It would read better to change this approach, to reflect a more qualitative ethos.

We revised the manuscript as per the reviewer’s comment in the previous review regarding numerical reporting and removed numbers for themes and only reported on sub-themes to illustrate the emphasis placed on the various sub-themes by the participants. We hope that the reviewer will agree that our level of reporting is helpful and appropriate to the analysis and are happy to leave the final decision to the editor.

6) The section “psychosocial support” talks about where support was provided and not provided. However, the quotes only seem to present the positive aspects here (i.e. where support was evident). For example it is not clear from this quote (“All I needed was someone to lend me an ear, give me the space to break down.”) if the participants received support or not. Quotes that explicitly reflect both perspectives and experiences are warranted. (This is different to the following quote, where friends/family tried to offer support but just didn’t know how).

We have revised this section on page 8 and added the following quote to represent experiences of absence of support:

“It did not really help when my peers made no effort to show compassion towards me or understand what I was going through.” (Male, 25)

7) A more meaningful heading than “healthcare” is recommended.

“Healthcare” was suggested in place of “treatment” by the youth focus group (see “Patient and public involvement” on page 5) as they felt it was a more neutral term and encompassed various kinds of professional support. We hope that this clarifies the decision to use this heading.